# Variation in Glucose-6-Phosphate Dehydrogenase activity following acute malaria

**Benedikt Ley**[1]*, **Mohammad Shafiul Alam**[2], **Ari Winasti Satyagraha**[3], **Ching Swe Phru**[2], **Kamala Thriemer**[1], **Dagimawie Tadesse**[4], **Tamiru Shibiru**[4], **Asrat Hailu**[4], **Mohammad Golam Kibria**[2], **Mohammad Sharif Hossain**[2], **Hisni Rahmat**[3], **Jeanne R. Poespoprodjo**[5,6], **Wasif Ali Khan**[2], **Julie A. Simpson**[7], **Ric N. Price**[1,8,9]

**1** Global and Tropical Health Division, Menzies School of Health Research and Charles Darwin University, Darwin, Australia, **2** Infectious Diseases Division, International Centre for Diarrheal Diseases Research, Bangladesh, Mohakhali, Dhaka, Bangladesh, **3** Eijkman Institute for Molecular Biology, Jakarta, Indonesia, **4** Arba Minch University, College of Medicine & Health Sciences, Arba Minch, Ethiopia, **5** Timika Malaria Research Program, Papuan Health and Community Development Foundation, Timika, Papua, **6** Centre for Child Health-PRO, Faculty of Medicine, Public Health and Nursing, Universitas Gadjah Mada, Yogyakarta, Indonesia, **7** Centre for Epidemiology and Biostatistics, Melbourne School of Population and Global Health, University of Melbourne, Melbourne, Australia, **8** Centre for Tropical Medicine and Global Health, Nuffield Department of Clinical Medicine, University of Oxford, Oxford, United Kingdom, **9** Mahidol-Oxford Tropical Medicine Research Unit (MORU), Faculty of Tropical Medicine, Mahidol University, Bangkok, Thailand

* benedikt.ley@menzies.edu.au

**Data Availability Statement:** All relevant data are within the manuscript and its Supporting Information files.

**Funding:** This study was funded by the Bill & Melinda Gates Foundation (OPP1054404 and

## Abstract

Primaquine and tafenoquine are the only licensed drugs with activity against *Plasmodium vivax* hypnozoites but cause haemolysis in patients with glucose–6–phosphate dehydrogenase (G6PD) deficiency. Malaria also causes haemolysis, leading to the replacement of older erythrocytes with low G6PD activity by reticulocytes and young erythrocytes with higher activity. Aim of this study was to assess the impact of acute malaria on G6PD activity. Selected patients with uncomplicated malaria were recruited in Bangladesh (n = 87), Indonesia (n = 75), and Ethiopia (n = 173); G6PD activity was measured at the initial presentation with malaria and a median of 176 days later (range 140 to 998) in the absence of malaria. Among selected participants (deficient participants preferentially enrolled in Bangladesh but not at other sites) G6PD activity fell between malaria and follow up by 79.1% (95%CI: 40.4 to 117.8) in 6 participants classified as deficient (<30% activity), 43.7% (95%CI: 34.2 to 53.1) in 39 individuals with intermediate activity (30% to <70%), and by 4.5% (95%CI: 1.4 to 7.6) in 290 G6PD normal ($\geq$70%) participants. In Bangladesh and Indonesia G6PD activity was significantly higher during acute malaria than when the same individuals were retested during follow up (40.9% (95%CI: 33.4–48.1) and 7.4% (95%CI: 0.2 to 14.6) respectively), whereas in Ethiopia G6PD activity was 3.6% (95%CI: -1.0 to -6.1) lower during acute malaria. The change in G6PD activity was apparent in patients presenting with either *P. vivax* or *P. falciparum* infection. Overall, 66.7% (4/6) severely deficient participants and 87.2% (34/39) with intermediate deficiency had normal activities when presenting with malaria. These findings suggest that G6PD activity rises significantly and at clinically relevant levels during acute malaria. Prospective case-control studies are warranted to confirm

OPP1164105 awarded to RNP). KT is a CSL Centenary fellow, RNP is funded by the Wellcome Trust (Senior Fellowship in Clinical Science, 200909). icddr,b is also grateful to the Governments of Bangladesh, Canada, Sweden and the UK for providing core/unrestricted support. The funders had no role in study design, data collection and analysis, decision to publish, or preparation of the manuscript.

**Competing interests:** The authors have declared that no competing interests exist.

the degree to which the predicted population attributable risks of drug induced haemolysis is lower than would be predicted from cross sectional surveys.

## Author summary

*Plasmodium vivax* forms dormant liver stages in the human host that reactivate weeks to months after the first infection and cause significant morbidity and mortality in affected populations. The group of 8-aminoquinolines are the only class of licensed drugs that remove these liver stages from the human host but are contra-indicated in patients with low activities of the glucose-6-phosphate dehydrogenase enzyme (G6PD). The WHO therefore recommends testing G6PD activity prior to treatment to exclude individuals with low activities from standard treatment. We enrolled 335 patients with malaria in Bangladesh, Indonesia, and Ethiopia and measured G6PD activity in all participants. All participants were followed up and a second G6PD measurement was collected between 6 and 33 months after enrolment if participants were free of malaria. When comparing both measurements, G6PD activity was 10% higher during malaria. The increase in G6PD activity during malaria is probably triggered by the *Plasmodium* infection, the observed change in G6PD activity may alter the risk profile of standard malaria treatment with 8-aminoquinolines.

## Introduction

In Asia, Oceania, the Horn of Africa, and the Americas, *Plasmodium vivax* is the predominant cause of malaria, with an estimated 7–14 million cases of malaria each year [1,2]. *P. vivax* forms dormant liver stages (hypnozoites), that can reactivate weeks to months after the initial infection causing multiple relapses and febrile illness [3]. The only available compounds with activity against hypnozoites are primaquine (PQ) and tafenoquine (TQ) [4]. Both of these 8-aminoquinoline compounds cause oxidative stress that can result in severe haemolysis in patients with glucose-6-phosphate-dehydrogenase (G6PD) deficiency, an inherited enzymopathy, affecting approximately 400 million people globally [5,6].

G6PD is an essential enzyme in the pentose phosphate pathway, the only pathway for red blood cells (RBCs) to maintain their redox potential by reducing nicotinamide adenine dinucleotide phosphate (NADP+) to NADPH [7]. The *G6PD* gene is located on the X-chromosome, hence males are hemizygous and can be either G6PD deficient or normal, whereas females can be homozygous deficient, homozygous normal, or heterozygous for the gene [7]. The latter results in intermediate enzyme activity determined by the embryonic process of X-chromosome inactivation (lyonization) [5]. Mature RBCs do not have a nucleus and depend on the G6PD generated during erythropoiesis, hence reticulocytes and young RBCs have the greatest G6PD activity which decreases with the age of the RBC [7]. The risk of drug induced haemolysis is determined by the patients G6PD variant, the degree of enzyme activity, the degree of lyonization in females, the age of the red cell population, and the degree of exposure [8,9].

Malarial parasitaemia also causes haemolysis, arising from loss of both parasitized and unparasitized red blood cells [10,11]. In a recent study from Bangladesh G6PD activities differed significantly between patients with acute malaria and healthy individuals from the same area, whereas there was no significant difference in G6PD activity between individuals with

and without a history of malaria in a case control study [12]. These findings suggest that *Plasmodium spp.* parasitaemia affects G6PD activity independent of any protective effect of G6PD deficiency from acute malaria [12–15]. In this longitudinal study we have quantified the change in G6PD activity before and after an acute episode of malaria in the same individuals.

## Methods

### Ethics statement

Ethical approval was provided by the Human Research Ethics Committee of the Northern Territory, Australia (HREC), the Ethical Review Committee of the ICDDR, B (Bangladesh), the Medical and Health Research Ethics Committee (Indonesia), and the National Research Ethics Review Committee (Ethiopia). Written informed consent was collected from all participants prior to enrolment, in case of minors written informed consent was collected from a legal guardian and written informed assent was also collected in case the minor was above 11 years of age (S1 Table).

### Enrolment with malaria

Antimalarial clinical efficacy trials were conducted in Bandarban, eastern Bangladesh [16] (ClinicalTrials.gov Identifier: NCT02389374), Timika in southern Papua province, Indonesia (ClinicalTrials.gov Identifier: NCT02787070), and Arba Minch, in southern Ethiopia [17] (ClinicalTrials.gov Identifier: NCT01814683). At all three sites patients with uncomplicated malaria were enrolled following informed written consent from the patient or their legal guardian, pregnant and lactating women were excluded. In Indonesia and Bangladesh patients were infected with either *P. falciparum* or *P. vivax*, whereas in Ethiopia only patients with *P. vivax* were enrolled (S2 Table).

At enrolment a minimum of 3ml of venous blood were collected, hemoglobin (Hb) was measured, and malaria diagnosis was confirmed by blood film examination. All slides were read by two independent microscopists and the mean of both readings was recorded. Whenever discordant results were obtained, slides were re-read at a reference laboratory and the expert reading was considered as final. Remaining venous blood was stored in EDTA at 4–8˚C and transported to a reference laboratory within 48 hours [18]. Hb was again measured immediately prior to spectrophotometry in the laboratory (S3 Table).

Patients were treated according to study protocols and monitored daily until aparasitemic and then followed weekly for 6 weeks in Bangladesh, 6 months in Indonesia, and 12 months in Ethiopia (S2 Table).

### Repeat G6PD assessment during convalescence

A subset of patients enrolled into the original trials, who had been tested for G6PD activity, were contacted and those consenting to further investigation were enrolled into the longitudinal study. In Indonesia and Ethiopia all eligible individuals were approached. For logistical reasons, in Bangladesh a sample of 100 individuals (55%) with the lowest G6PD activity at baseline were selected, and patients who could not be contacted or declined to participate were replaced. Patients consenting to further investigation had a venous blood sample collected for repeat blood film examination or rapid diagnostic test for malaria and quantitative G6PD assay. Patients with peripheral parasitaemia by microscopy or RDT as well as patients with self-reported history of malaria within the last 90 days, or major blood loss were excluded (S3 Table and S1 Fig).

## Assessment of G6PD activity

G6PD activity was measured on temperature-controlled spectrophotometers. At each site the same equipment and assay method was applied to assess the same individuals during an acute episode of malaria and again during follow up. In Bangladesh and Indonesia, measurements were done on a Shimadzu 1800 (Shimadzu, Japan) spectrophotometer and in Ethiopia a Humalyzer 3000 (Human, Germany) was used. Assay kits were supplied by Randox Laboratories (PD410, UK) in Bangladesh, and by Trinity Biotech (345-A, Ireland) in Indonesia and Ethiopia. Each measurement was done in duplicate, and whenever these differed by more than 15% a third measurement was undertaken. If all 3 measurements differed by more than 15% the sample was re-tested in duplicate. All G6PD results were normalized by Hb measured directly prior to spectrophotometry using a Hemocue 301, (Hemocue, Sweden). Other laboratory methods and follow up periods are presented in S3 Table.

G6PD normal and G6PD deficient controls were run throughout the clinical trial in Bangladesh (September 2014 until March 2015) and again during the period of convalescence review (March 2017 to May 2017). In Indonesia and Ethiopia G6PD normal, intermediate, and deficient controls were run in regular intervals throughout the duration of the study (October 2016 to May 2018 and January 2017 to October 2017, respectively) (S2 Fig and S4 Table).

## Data management and statistical analysis

Data collected at enrolment were extracted from existing databases, including sex, age, G6PD activity, Hb concentration (at baseline and day 7), and species of infection.

The primary endpoint was G6PD activity measured during acute malaria and follow up. To pool enzyme activity across sites, individual G6PD activities were normalized and expressed as a fraction of 100% G6PD activity, defined as the adjusted male median in the local population (AMM) [19]. Since participants in Bangladesh were selected purposively 100% G6PD activity was defined as 7.03U/gHb based on results from a previous cross-sectional survey conducted in the same area using the same laboratory methods [20]. In Indonesia and Ethiopia 100% G6PD activity was defined as the AMM based on the follow up measurements. G6PD activity was categorized using the following cutoff values: <30% activity was defined as G6PD deficient, ≥30% to <70% as intermediate activity and ≥70% activity as G6PD normal.

The change in normalized G6PD activity (quantified as a percentage of the AMM) was analyzed using a linear mixed-effects model, with G6PD activity as the dependent variable, the categorical time point of measurement (malaria or follow up) and study site (Bangladesh, Ethiopia, and Indonesia) as fixed effects and the individual participant as random effect. The analysis was repeated stratifying by the species of initial infection. Since G6PD activity is normalized by Hb, changes in activity are vulnerable to associated changes in Hb concentration. Analysis was therefore repeated adjusting for the time-varying covariate, Hb concentrations (in g/dL), measured at the malaria episode and follow-up.

To assess whether the observed difference in activities was due to effects from regression to the mean (rtm), rtm effects were calculated considering 30% and 70% activity as cut-offs and the follow up measurement when participants were aparasitemic as baseline activity. The calculated rtm effect was then compared to the observed effect [21].

To assess within-individual agreement in classification of deficient, intermediate, and normal G6PD activity using measurements taken at time of malaria and follow up, the extended McNemmars test for correlated proportions was performed on the whole study population and stratified by species.

It was assumed that in the absence of haemolysis or major blood loss, G6PD activity does not vary significantly over time, and therefore G6PD activity during follow up reflects an

individual's G6PD activity at steady state. As a proxy for drug induced haemolysis the change in Hb between the day when Pq radical cure was first administered and seven days later was considered. Linear regression analysis was used to determine whether activity during steady state or the G6PD activity during the malaria episode was predictive of drug induced haemolysis. Hb concentration on day 7 was the dependent variable and G6PD activity during malaria and follow up were considered exposures after controlling for sex, parasite density, and Hb at treatment start [10,22]. Since day 7 Hb was not recorded for Indonesian patients this cohort was excluded from this analysis.

## Results

Overall, 335 participants were included in the paired analysis, representing 27.4% of the 1,221 patients enrolled into the original clinical trials. In Bangladesh out of the planned 100 individuals 87 patients could be contacted and their identity verified (26.0%), while 75 (22.4%) patients from Indonesia and 173 (51.6%) patients from Ethiopia were included. In Bangladesh 33.3% (29/87) had a *P. vivax* infection, in Indonesia 49.3% (37/75) had a *P. vivax* infection and in Ethiopia only patients with *P. vivax* infection were enrolled (Table 1). The baseline characteristics were similar between the patients included in the analysis and the overall cohort, except for Hb readings in Indonesia and parasite density in Ethiopia.

The median time between presentation with malaria and follow up varied significantly between sites: 887 days in Bangladesh (range: 790 to 998), 168 days in Indonesia (range: 140–168), and 176 days in Ethiopia (range: 169–183) (S2 Fig). Overall, the mean Hb concentration was lower at acute presentation compared to follow up (mean difference: -0.9 g/dL, 95% confidence interval [95%CI]: -1.1 to -0.6, p<0.001). This trend was apparent for patients enrolled in Bangladesh (-1.3 g/dL, 95%CI: -1.7 to -0.9, p<0.001) and Ethiopia (-1.1 g/dL, 95%CI:-1.3 to -0.9, p<0.001), but was not significant in Indonesia (0.2 g/dL, 95%CI: -0.5 to 0.9, p = 0.555).

**Table 1. Characteristics of selected and non-selected participants per study site.**

| | Bangladesh | | Indonesia | | Ethiopia | |
|---|---|---|---|---|---|---|
| | Excluded cohort | Selected Cohort | Excluded cohort | Selected Cohort | Excluded cohort | Selected Cohort |
| Number | 94 | 87 | 591* | 75 | 201 | 173 |
| Females (%) | 21 (22.3) | 25 (28.7) | 312 (52.8) | 35 (46.7) | 98 (49.5) | 78 (45.1) |
| *P.falciparum* (%) | 61 (64.9) | 54 (62.1) | 313 (52.3) | 38 (50.7) | 0 (0.0) | 0 (0.0) |
| *P.vivax* (%) | 26 (27.7) | 29 (33.3) | 261 (46.7) | 37 (49.3) | 198 (100.0) | 173 (100.0) |
| Mixed infection (%) | 7 (7.4) | 4 (4.6) | 24 (4.0) | 0 (0.0) | 0 (0.0) | 0 (0.0) |
| Geometric mean (95% of range) parasite density for *P. falciparum* and mixed infections | 3,924 (80–80,000) | 8,903 (80–960,000) | 4,532 (200–76,000) | 5,314 (360–84,000) | - | - |
| Geometric mean parasite density (per µl) (95% of range) for *P.vivax* | 2,920 (360–26,600) | 2,507 (80–28,720) | 3,796 (160–36,000) | 2,236 (80–18,440) | 15,433 (889–167,500) | 9,143 (667–117,500) |
| Geometric mean parasite density (per µl) (95% of range) for mixed infections | 6,124 (200–460,000) | 24,878 (667–185,000) | 894 (0–1,040) | 0 (0–0) | - | - |
| Mean Hb in g/dL (95%CI) on day of enrolment | 12.5 (12.1–12.9) | 12.8 (12.3–13.3) | 11.9 (11.8–12.1) | 12.6 (11.9–13.2) | 13.2 (13.0–13.4) | 13.3 (13.0–13.4) |
| Median G6PD activity at enrolment in U/gHb (IQR) | 8.8 (8.3–9.5) | 7.5 (6.2–9.5) | - | 10.3 (9.7 to 10.9) | 10.8 (9.3–12.2) | 11.5 (10.6 to 12.8) |
| 100% G6PD activity (in U/gHb)** | 7.03 | | 9.24 | | 12.13 | |

*the clinical trial comprised of 420 participants and was complemented by an observational arm, G6PD activity was only measured in the selected cohort

** from literature in Bangladesh [23] and calculated from convalescent subset in Indonesia and Ethiopia

## Changes in acute and convalescent G6PD activity

Based on results from a previous cross sectional study in the same population 100% G6PD enzyme activity was defined as 7.03 U/gHb in Bangladesh [23]; based on the G6PD measurements during follow up 100% activity was defined as 9.24U/gHb in Indonesia, and 12.13 U/gHb in Ethiopia (Table 1). When comparing the same individual's G6PD activity during malaria and again during follow up, enzyme activity was -10.4% (95% confidence interval [95%CI]: -13.9 to -6.9) lower during follow up. The decrease in activity from malaria to follow-up was greatest in those classified as G6PD deficient at follow-up (<30% activity) (-79.1%, 95%CI: -117.8 to -40.4), compared to -43.7% (95%CI: -53.1 to -34.2) in those with intermediate deficiency at follow-up (≥30% to <70% activity), and -4.5% (95%CI: -7.6 to -1.4) in those who were G6PD normal at follow-up (≥70% activity) (Fig 1).

Only one male patient with *P. vivax* mixed infection from Bangladesh was deficient and his activity did not change between enrolment and follow up (0.7U/gHb at both time points), while activity decreased by -95.0% (95%CI: -126.5 to—63.5) in the 5 deficient patients with *P. falciparum* infection. Among the 17 intermediate patients with *P. vivax* mono or mixed infection, activity was -49.4% (95%CI: -61.1 to -37.8) lower at the time of follow-up, while activity dropped by -39.2% (95%CI: -52.8 to -25.6) in the 22 intermediate patients with *P. falciparum* mono-infection (Table 2).

The calculated rtm effect for deficient individuals was -60.9% (95%CI: –71.4 to -50.5, p<0.001) at the 30% activity threshold and -31.4% (95%CI: -36.8 to -25.9, p<0.001) at the 70% threshold.

The greatest mean difference between initial presentation with malaria and follow up was observed in Bangladesh (-40.9%, 95%CI: -48.1 to –33.4), followed by Indonesia (-7.4%, 95%CI: -14.6 to -0.2), while in Ethiopia G6PD activity was slightly lower during acute malaria compared to follow up (3.6%, 95%CI: 1.0 to 6.1); Fig 2.

The country specific drop in activity corresponded to the proportion of participants with decreased activities enrolled. In Bangladesh 33 (37.9%) individuals were categorized with intermediate G6PD deficiency and 6 (6.9%) with severe deficiency. The corresponding proportions were significantly lower in Indonesia (6.7% and 0.0%) and Ethiopia (0.6% and 0.0% respectively). The proportions of participants categorized as normal, intermediate, and deficient varied significantly between time points (p<0.001). Overall, 4 of the 6 participants (66.7%) classified as G6PD deficient during the follow up were categorized as G6PD normal

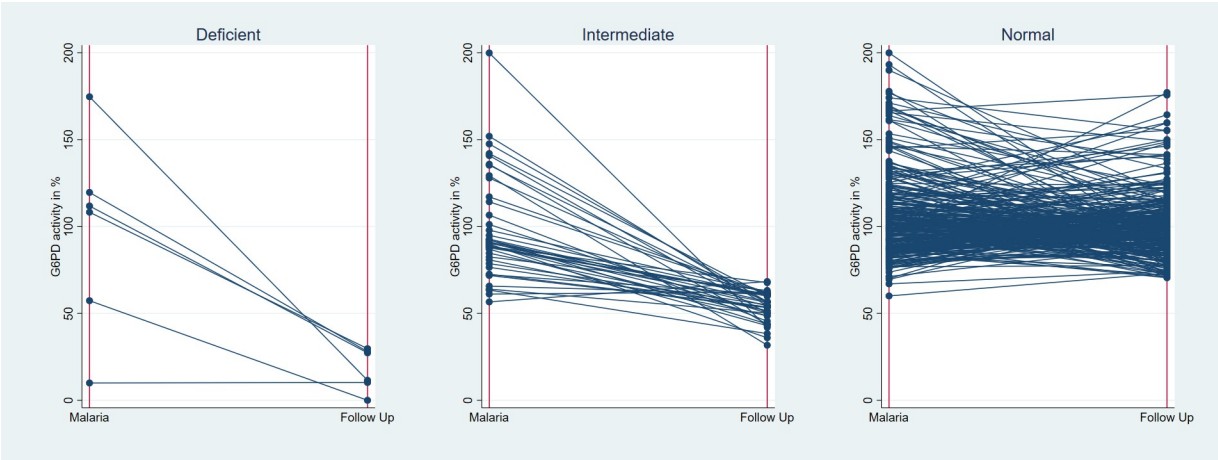

**Fig 1. Change in G6PD activity between the malaria episode and follow up, stratified by G6PD status at the time of follow up.**

**Table 2. Percentage Change (95% Confidence Intervals) in G6PD activity between acute malaria and follow up.**

| Sites | Species | Deficient* | Intermediate* | Normal* |
|---|---|---|---|---|
| **All Countries** | **All species (95%CI, n):** | -79.1 (-117.8 to -40.4, 6) | -43.7 (-53.1 to -34.2, 39) | -4.5 (-7.6 to -1.4, 290) |
| | *P. vivax* **mono and mixed infections (95%CI, n)** | -0.3 (n = 1) | -49.4 (-61.1 to -37.8, 17) | -1.4 (-4.6 to 1.7, 225) |
| | *P. falciparum* **mono-infection (95%CI, n)** | -95.0 (-126.5 to -63.5, 5) | -39.2 (-52.8 to -25.6, 22) | -15.2 (-23.0 to -7.3, 65) |
| **Bangladesh** | **All species (95%CI, n)** | -79.1 (-117.8 to -40.4, 6) | -48.5 (-58.9 to -38.0, 33) | -30.7 (-39.1 to -22.3, 48) |
| | *P. vivax* **mono and mixed infections (95%CI, n)** | 0.3 (n = 1) | -56.3 (-70.2 to -42.3, 14) | -34.1 (-48.6 to -19.6, 18) |
| | *P. falciparum* **mono-infection (95%CI, n)** | -95.0 (-144.9 to -45.1, 5) | -42.7 (-60.7 to -24.7, 19) | -28.7 (-40.2 to -17.2, 30) |
| **Ethiopia** | **All species (95%CI, n)** | NA (n = 0) | 1.2 (n = 1) | 3.6 (1.0 to 6.1, 172) |
| | *P. vivax* **mono and mixed infections (95%CI, n)** | NA (n = 0) | 1.2 (n = 1) | 3.6 (1.0 to 6.1, 171) |
| | *P. falciparum* **mono-infection (95%CI, n)** | NA (n = 0) | NA (n = 0) | NA (n = 0) |
| **Indonesia** | **All species (95%CI, n)** | NA (n = 0) | -21.0 (-36.4 to -5.5, 5) | -6.4 (-14.0 to 1.2, 70) |
| | *P. vivax* **mono and mixed infections (95%CI, n)** | NA (n = 0) | -26.7 (-34.2 **and** -19.3, 2) | -9.2 (-21.4 to 3.0, 35) |
| | *P. falciparum* **mono-infection (95%CI, n)** | NA (n = 0) | -17.0 (-37.6 **and** -25.0 **and** 11.6, 3) | -3.6 (-13.6 to 6.5, 35) |

*Defined during follow up

and 1 (16.7%) as G6PD intermediate at time of their acute presentation. Among 39 individuals classified as having intermediate activity, 34 (87.2%) were categorized as G6PD normal during acute malaria, whereas none were categorized as G6PD deficient. Conversely of the 290 individuals categorized as G6PD normal during follow up, 3 (1.0%) participants were classified as G6PD intermediate during the malaria episode. All 3 participants were enrolled in Indonesia and had between 60% and 70% activity during enrolment. The shift in categories was significant in individuals presenting with *P. vivax* and *P. falciparum* infection and in Bangladesh (Table 3).

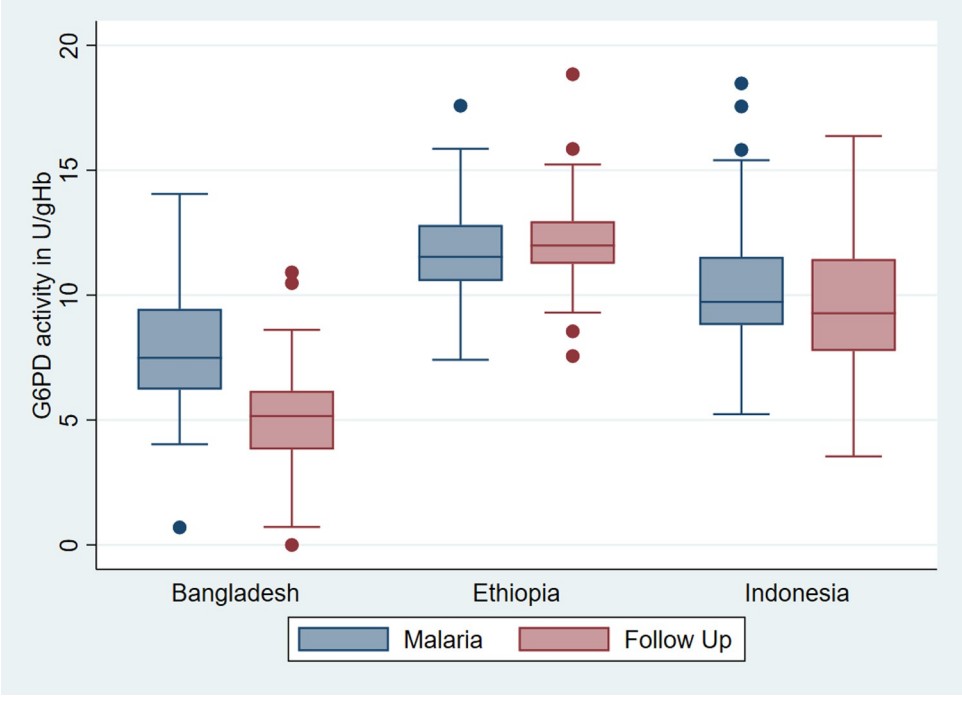

**Fig 2. Box and Whiskers plot of median normalized G6PD activity during malaria and follow up per study site among selected participants.**

**Table 3. Participants categorized by G6PD status during acute malaria and follow up stratified by species and site.**

| | | | G6PD activity during follow up | | | |
| --- | --- | --- | --- | --- | --- | --- |
| | | | Deficient (%) | Intermediate (%) | Normal (%) | Total (%) |
| **G6PD activity during acute malaria** | **All species, n = 347** | **Deficient** | 1 (16.7%) | 0 (0.0%) | 0 (0.0%) | **1 (0.3)** |
| | | **Intermediate** | 1 (16.7%) | 5 (12.8%) | 3 (1.0%) | **9 (2.7)** |
| | | **Normal** | 4 (66.7%) | 34 (87.2%) | 287 (99.0%) | **325 (97.0)** |
| | | **Total (%)** | **6 (1.8)** | **39 (11.6)** | **290 (86.6)** | **335 (100.0, p<0.001)** |
| | ***P. vivax* mono and mixed infections, n = 247** | **Deficient** | 1 (100.0%) | 0 (0.0%) | 0 (0.0%) | **1 (0.4)** |
| | | **Intermediate** | 0 (0.0%) | 1 (5.9%) | 3 (1.3%) | **4 (1.7)** |
| | | **Normal** | 0 (0.0%) | 16 (94.1%) | 222 (98.7%) | **238 (97.9)** |
| | | **Total (%)** | **1 (0.4)** | **17 (7.0)** | **225 (92.6)** | **243 (100.0, p = 0.003)** |
| | ***P. falciparum* mono-infection, n = 100** | **Deficient** | 0 (0.0%) | 0 (0.0%) | 0 (0.0%) | **0 (0.0)** |
| | | **Intermediate** | 1 (20.0%) | 4 (18.2%) | 0 (0.0%) | **5 (5.0)** |
| | | **Normal** | 4 (80.0%) | 18 (81.8%) | 65 (100.0%) | **95 (95.0)** |
| | | **Total (%)** | **5 (5.4)** | **22 (23.9)** | **65 (70.7)** | **92 (100.0, p<0.001)** |
| | **Bangladesh, n = 87** | **Deficient** | 1 (100.0) | 0 (0.0) | 0 (0.0) | **1 (1.2)** |
| | | **Intermediate** | 1 (25.0) | 2 (50.0) | 1 (25.0) | **4 (4.6)** |
| | | **Normal** | 4 (4.9) | 31 (37.8) | 47 (57.3) | **82 (94.3)** |
| | | **Total (%)** | **6 (6.9)** | **33 (37.9)** | **48 (55.2)** | **87 (100.0, p<0.001)** |
| | **Ethiopia, n = 172** | **Deficient** | 0 (0.0) | 0 (0.0) | 0 (0.0) | **0 (0.0)** |
| | | **Intermediate** | 0 (0.0) | 1 (100.0) | 0 (0.0) | **1 (0.6)** |
| | | **Normal** | 0 (0.0) | 0 (0.0) | 172 (100.0) | **172 (99.4)** |
| | | **Total (%)** | **0 (0.0)** | **1 (0.6)** | **172 (99.4)** | **173 (100.0, p = 1.00)** |
| | **Indonesia, n = 75** | **Deficient** | 0 (0.0) | 0 (0.0) | 0 (0.0) | **0 (0.0)** |
| | | **Intermediate** | 0 (0.0) | 2 (50.0) | 2 (50.0) | **4 (5.3)** |
| | | **Normal** | 0 (0.0) | 3 (4.2) | 68 (95.8) | **71 (94.7)** |
| | | **Total (%)** | **0 (0.0)** | **5 (6.7)** | **70 (93.3)** | **75 (100.0, p = 0.655)** |

Deficient: <30% G6PD activity, Intermediate: ≥30% to <70% G6PD activity, Normal: ≥70% G6PD activity

In total 30 patients from Bangladesh and 172 patients from Ethiopia had a Hb concentration measured at enrolment and seven days after starting PQ based radical cure. Their Hb dropped by a mean of 0.1 g/dL (95%CI: -0.3 to 0.0) in the first seven days of treatment. A 10% increase in G6PD activity at enrolment was associated with a drop in day 7 mean Hb of -0.04 g/dL (95%CI: -0.13 to 0.06, p = 0.471) and a 10% increase in G6PD activity during steady state was associated with a decrease of 0.02 g/dL (95%CI: -0.11 to 0.07, p = 0.611).

## Discussion

Our study highlights that G6PD activity in the same individual can differ significantly when measured during acute presentation with malaria and after recovery. Overall an individual's G6PD activity was 10% greater when measured during an acute episode of malaria compared to when remeasured during follow up. When G6PD status was categorized according to an individual's enzyme activity during follow up, two thirds of participants with severe deficiency and almost 90% of those with intermediate deficiency had normal enzyme activity (>70%) at the time of presentation with malaria. The overall change in activity was more pronounced for patients with *P. vivax* infection, however the differences in an individual's G6PD status were apparent in patients presenting with *P. vivax* and those presenting with *P. falciparum*

infection. Among selected participants the greatest change in G6PD activity was observed in Bangladesh where a large proportion of deficient and intermediate individuals were included, followed by Indonesia where two out four participants with intermediate activities had normal activities when revisited. In Ethiopia only one individual had reduced G6PD activity during follow up, a 25-year-old male with 61% enzyme activity during malaria and 62% during follow up. Whilst the overall mean G6PD activity in Ethiopian participants was 4% higher during follow up compared to during the malaria episode, this modest rise was not clinically relevant. This is also reflected when considering the change in in G6PD category between enrolment and follow up. There was a significant difference in participants G6PD status between enrolment and follow up, however this only reached statistical significance in Bangladesh where 45% of participants were G6PD deficient during follow up. In contrast only 1% of participants in Ethiopia and 7% in Indonesia had enzyme activities <70% during follow up.

G6PD is essential for the production of glutathione, which binds free radicals and protects cells from oxidative stress [7,9,24]. Since mature RBCs do not have a nucleus, they depend on the G6PD synthesised during erythropoiesis, hence enzyme activity is highest in reticulocytes and young RBCs, and subsequently falls as erythrocytes age [7]. During an acute *P.vivax* or *P. falciparum* infection, loss of RBCs occurs from haemolysis of both parasitized and unparasitized erythrocytes [10,25–27]. The ratio of infected to uninfected RBCs lost is estimated at 1:34 for *P.vivax* and 1:8 for *P. falciparum* [28]. While the underlying mechanisms for the loss in unparasitized RBCs is not fully understood [29], it has been attributed to oxidative membrane damage induced by the malaria parasite [30]. G6PD plays a crucial role in maintaining RBC redox potential and reducing oxidative stress, thus as G6PD activity decreases with RBC age, older cells are preferentially removed from the peripheral blood [8]. Furthermore parasite induced hemolysis results in a decrease in Hb concentration and oxygen delivery that triggers reactive erythropoiesis and production of young RBCs with higher G6PD activity [10], hence the mean age of the hosts RBC population decreases and the overall G6PD activity increases.

The observation that G6PD activity increases during acute malaria is not unexpected, however the degree to which this occurred in our study was surprising with potentially important public health implications. The prevalence of G6PD deficiency is highest in *P. vivax* endemic areas and can result in healthcare providers being reluctant to prescribe radical cure due to fears of drug induced haemolysis [31,32]. Our findings suggest that in patients presenting with *P. vivax* malaria the prevalence of severe and intermediate G6PD deficiency is considerably lower than that predicted from cross sectional surveys; if confirmed these differences will influence population-based risk and cost -benefit estimates for different test and treat strategies for primaquine radical cure [33]. Our findings also support previous models proposing a delay in starting hypnozoitocidal treatment start until the end of schizontocidal treatment (once the most deficient RBCs have been lysed), to reduce drug induced haemolysis [34].

We were unable to determine whether G6PD activity during malaria or follow up was predictive of haemolytic risk following PQ treatment, since only two (1.0%) out of the 202 participants with Hb measured on baseline and day 7 who received 14-day primaquine had intermediate activities during malaria and none were deficient. The degree to which transient increases in G6PD activity alter the risks of drug induced haemolysis will require further prospective clinical studies.

Current WHO guidelines recommend testing for G6PD deficiency prior to starting primaquine so that individuals at greatest risk of haemolysis can be given alternative treatment [35]. Our findings raise the possibility that aparasitemic individuals identified as severely or intermediate G6PD deficient may not necessarily be at high risk of drug induced haemolysis during malaria. A total of 34 out of 39 participants classified as intermediate deficient when

aparasitemic had normal activities when enrolled with malaria, 4 out of 6 participants diagnosed as deficient when aparasitemic were G6PD normal when tested during acute malaria. Conversely 287 out of 290 individuals categorized as G6PD normal during follow up had no change in G6PD status when presenting with malaria. All three participants with normal G6PD activities during malaria and intermediate activities during follow up, had borderline G6PD activity (60% to 70%) at the first measurement and the mean increase in G6PD activity during follow up was less than 20%.

In a recent study from Cambodia, G6PD activity was measured over an 8-week period in patients with *P. vivax* malaria treated with weekly PQ [36]. G6PD activity increased between the first and second PQ dose, with the greatest change in activity observed in deficient patients. G6PD activity did not change markedly throughout the rest of the treatment period irrespective of G6PD status, likely due to the weekly PQ doses continuously hemolyzing RBCs with low G6PD levels [37]. G6PD activity was not measured again following completion of treatment and subsequent return to steady state, so the baseline normal state of these participants could not be determined.

In our study 12% of males had intermediate G6PD activity (30%-70%) recorded during follow up (S5 Table); this was most apparent in Bangladesh where patients with lowest G6PD activities were purposively selected. The G6PD gene is located on the X-chromosome and males are thus either hemizygous deficient or normal [8], however intermediate activities have been reported from previous cross-sectional surveys [23,38,39] and may reflect the AMM (100% activity) being calculated from heterogenous populations with G6PD variants associated with different severity.

Our study has several important limitations. First, the study adopted an opportunistic design, revisiting individuals who had been enrolled into previous antimalarial clinical trials. Since only patients initially presenting with malaria were enrolled and there were no aparasitemic controls, one cannot definitively infer that initial elevations in G6PD activity were attributable to the initial peripheral parasitaemia. Spectrophotometry procedures at the study sites could have changed, systematically altering either rounds of G6PD measurement which in Bangladesh occurred more than 2 years after the initial presentation [40]. Reassuringly quality control procedures suggested consistent testing throughout the study period at all three sites. There was a small but significant trend in deficient control isolates in Indonesia, but this would not have affected the outcome since none of the patients enrolled were severely deficient at either time point. Secondly, G6PD activity was analyzed after normalizing for Hb concentration, hence changes in the Hb between the first and second measurement could have impacted the observed changes in G6PD activity. However, after adjusting for Hb concentration in the multivariable analysis the estimated changes in G6PD activity levels between the acute malaria episode and follow-up were still apparent. Thirdly, the time to follow up differed significantly between sites. Haematological recovery following acute malaria usually occurs within 28–63 days [41,42], although after haemolytic events it can take significantly longer for the age distribution of RBC to return to steady state [43]. The primary objective of the current study was to resample patients following complete haematological recovery and during a period when the patients were aparasitaemic with no recent history of major blood loss or malaria within the previous 3 months, and this was achieved in all study sites with a minimum time to resampling of 140 days.

Fourth our analysis did not include genetic confirmation of the diagnosis of individuals with G6PD deficiency. More than 230 clinically relevant G6PD variants have been described [8,44] that confer different degrees of enzymatic deficiency. The observed changes in enzyme activity associated with acute haemolysis may differ between G6PD variants. For example, for individuals with the Mediterranean variant almost all RBCs, including young cells, are severely

deficient and thus parasite or drug induced haemolysis will have a minimal impact on the overall activity, whereas for individuals with milder variants, G6PD activity would be more likely to normalize. [45,46]. Hence observed changes in enzyme activity may differ significantly by region.

Finally, the observed difference in activities could have arisen from regression to the mean, with extreme values measured during follow up becoming less extreme when measured again during acute malaria [21]. However, at both the 30% and 70% definitions for G6PD deficiency the calculated rtm effect was below the observed effect. Keeping the small sample size of deficient individuals in mind, this suggests a true trend.

Our study is confounded by heterogeneity between study sites, selection bias (specifically in Bangladesh), differences in prevalence of G6PD deficiency, potentially known local variants, and potential regression to the mean. Despite these limitations, the changes in G6PD activity were apparent in varying degrees across all three sites and highlight that a substantial proportion of individuals with severe and intermediate G6PD deficiency were diagnosed as G6PD normal when presenting with acute malaria. Subsequent prospective case-control studies, with molecular confirmation of genetic variants, are warranted to confirm and quantify these findings in other endemic settings.

In conclusion, our study highlights that an individual's G6PD activity is significantly higher during acute malaria, particularly those with intermediate or severe G6PD deficiency. If these findings are confirmed, it raises the possibility that an individuals' G6PD status is not static, and thus the risk of drug induced haemolysis when presenting with malaria cannot be assumed from measuring an individual's enzyme activity when aparasitaemic. From a public health perspective, the population attributable risks of drug induced haemolysis may be significantly lower than that predicted from cross sectional surveys.

## Supporting information

**S1 Table. Ethical boards and protocol numbers.**
(DOCX)

**S2 Table. Study details on studies from which patients were recruited.** * This trial is complemented by an observational arm with a total sample size of 420 that is not registered with Clinicaltrials.gov. *** Not applicable for complementary arm **this site is part of a multi-center trial, however measurements relevant to this study were only performed at this site.
(DOCX)

**S3 Table. Site specific procedures.**
(DOCX)

**S4 Table. Quality control for spectrophotometry.**
(DOCX)

**S5 Table. Sex distribution per G6PD status based on G6PD activity during follow up.**
(DOCX)

**S1 Fig. Consort chart.**
(TIF)

**S2 Fig. Results of quality control testing througout the study period at the study sites.** Legend: top horizontal line: line of best fit for G6PD normal controls, center horizontal line: line of best fit for G6PD intermediate controls (not done in Bangladesh), lowest horizontal line: line of best fit for G6PD deficient controls; vertical line (Bangladesh) follow up period where

no testing was done
(TIF)

**S3 Fig. Days between enrolment and follow up.**
(TIF)

**S1 File. Underlying database.**
(XLS)

## Acknowledgments

We thank all study participants and study staff involved in the included studies.

## Author Contributions

**Conceptualization:** Benedikt Ley, Tamiru Shibiru.

**Data curation:** Benedikt Ley, Mohammad Sharif Hossain.

**Formal analysis:** Benedikt Ley, Julie A. Simpson, Ric N. Price.

**Funding acquisition:** Ric N. Price.

**Investigation:** Benedikt Ley, Mohammad Shafiul Alam, Ari Winasti Satyagraha, Ching Swe Phru, Dagimawie Tadesse, Mohammad Golam Kibria, Hisni Rahmat.

**Methodology:** Benedikt Ley.

**Project administration:** Mohammad Shafiul Alam.

**Validation:** Benedikt Ley.

**Writing – original draft:** Benedikt Ley.

**Writing – review & editing:** Benedikt Ley, Mohammad Shafiul Alam, Ari Winasti Satyagraha, Ching Swe Phru, Kamala Thriemer, Dagimawie Tadesse, Tamiru Shibiru, Asrat Hailu, Mohammad Sharif Hossain, Jeanne R. Poespoprodjo, Wasif Ali Khan, Julie A. Simpson, Ric N. Price.

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
