## [Decision Letter · Decision Letter 0]

1 Dec 2021

Dear Dr Ley,

Thank you very much for submitting your manuscript "Variation in Glucose-6-Phosphate Dehydrogenase activity following acute malaria" for consideration at PLOS Neglected Tropical Diseases. As with all papers reviewed by the journal, your manuscript was reviewed by members of the editorial board and by several independent reviewers. In light of the reviews (below this email), we would like to invite the resubmission of a significantly-revised version that takes into account the reviewers' comments. 

We cannot make any decision about publication until we have seen the revised manuscript and your response to the reviewers' comments. Your revised manuscript is also likely to be sent to reviewers for further evaluation.

Sincerely,

Hans-Peter Fuehrer

Deputy Editor

Hans-Peter Fuehrer

Deputy Editor

Reviewer's Responses to Questions

**Key Review Criteria Required for Acceptance?**

**Methods**

-Are the objectives of the study clearly articulated with a clear testable hypothesis stated?

-Is the study design appropriate to address the stated objectives?

-Is the population clearly described and appropriate for the hypothesis being tested?

-Is the sample size sufficient to ensure adequate power to address the hypothesis being tested?

-Were correct statistical analysis used to support conclusions?

-Are there concerns about ethical or regulatory requirements being met?

Reviewer #1: “A subset of patients enrolled in the original trials were reviewed again 6 to 33 months after enrolment.” This statement is rather vague. Please indicate clearly how participants in the study were selected. Please provide a “consort chart” illustrating how the study participants were assembled? It may be necessary to provide separate charts for each site.

Please state clearly inclusion/exclusion criteria? If the criteria differ between sites, please provide a table with the criteria for each site?

“Since participants in Bangladesh were selected purposively …” I can’t follow this argument – please explain?

Reviewer #2: Methods; line 116: clarify better for the different studies how the subset were selected for the follow-up. In Ethiopia/Indonesia, was it just based on participant contactability? This matters given importance of the AMM estimates on the results. Also please clearly state the participants’ health status at the 2nd time point.

To avoid misinterpretation, it would be good to emphasise that the rates of deficiency seen at the two time points are based on selected sample groups: “among selected patients” (only G6PD normal indivs recruited in Ethiopia/Indonesia at first timepoint; G6PD deficient patients preferentially included in Bangladesh). E.g. line 57 in Abstract, Fig 1 legend, Line 243 “selectively enrolled”.

Reviewer #3: The design is not ideal to address the stated objectives as there are no control cases in the study.

The statistics presented are misleading and in some places inappropriately used to substantiate the authors conclusions

**Results**

-Does the analysis presented match the analysis plan?

-Are the results clearly and completely presented?

-Are the figures (Tables, Images) of sufficient quality for clarity?

Reviewer #1: Table 1: please explain what you mean by “unselected cohort”? I assume you mean excluded? Please highlight the this excluded cohort in the consort chart?

“The time between presentation with malaria and follow up ranged between 790 and 998 days in Bangladesh, 140 and 168 days in Indonesia, and 169 and 183 days in Ethiopia.” Please provide a more detailed statistical description of the distribution of F/U dates. Pls. consider providing a Figure to illustrate the distributions.

“The greatest difference between acute presentation and follow up was observed in Bangladesh” not only was this difference greatest in Bangladesh. The time window between 1st and 2nd measurement was also by the widest. Perhaps the extended duration explains the differences in enzyme activity? Pls. explain why you think this is not the case?

“Figure 2: Change in G6PD activity between malaria and follow up, stratified by G6PD status during follow up” Pls. rephrase the title? … between the malaria episode and follow up, stratified by G6PD status at the time of follow up

Pls consider a similar analysis as presented in Fig 2 instead of stratified G6D status stratified by site? (at a minimum include table S5 in the main text.)

“The estimated mean change in Hb associated with an increase of 10 % units in G6PD activity was -0.05 g/dL (95% CI: -0.14 to 0.05, p=0.336) at enrolment and -0.03 g/dL (95% CI: -0.11 to 0.06, p=0.560) at follow up.” This observation could be interesting but seems to be difficult to understand the way it is presented. The change in enzyme activity is over the time period from malaria episode to F/U date? Why don’t you report the Hb change over the same period?

Reviewer #2: The implications of the results are most significant for P. vivax patients. Would there be statistical power to allow a subanalysis with only Pv patients?

Supplementary figure 1: suggest including control recommended ranges to help interpret the variation seen. Could also add control manufacturer names to plot titles.

Reviewer #3: The results are clearly presented but the analysis is misleading

A major limitation in the analysis is the fact that the analysis (change in G6PD activity) per Plasmodium species is not presented, and the fact that the one site with only P.vivax infections does not support their conclusions is not discussed.

**Conclusions**

-Are the conclusions supported by the data presented?

-Are the limitations of analysis clearly described?

-Do the authors discuss how these data can be helpful to advance our understanding of the topic under study?

-Is public health relevance addressed?

Reviewer #1: Discussion

“Activity increased by 11% overall, ranging by almost 80% in deficient individuals, 45% in

intermediate individuals, and 5% in individuals categorized as G6PD normal during follow up.” This sentence can be interpreted in different ways. Either the enzyme activity increased during dollow up or the individuals were categorized as normal during follow up. Please consider rewriting to provide more clarity?

“during health” is poor English – pls. rephrase?

“Over 80% of study participants diagnosed with less than 70% activity during follow up would have

been denied primaquine radical cure if they had not been re-tested.” In this sentence the temporal relationship between the first test and the second test is the wrong way around. Fewer readers may be confused if you would consider rephrasing? “Over 80% of study participants diagnosed with less than 70% enzyme activity during follow up would have been denied the radical cure if health care providers would rely solely on the follow-up test instead of the test result at the time of the acute malaria episode”.

“Reassuringly quality control procedures suggested reliable testing throughout the study period at all three sites. ‘ perhaps the keyword here is “consistent” rather than “reliable”?

Please discuss the advantages/disadvantages of combining data from different sites when the data are so different. Isn’t that comparing “apples with oranges”?

Conclusions:

“Our findings have important public health implications for identifying populations at risk of haemolysis, as well as test and treat strategies for radical cure.” This sentence is meaningless – why waste the time of the reader? Please clearly state the implications!

Reviewer #2: (No Response)

Reviewer #3: The conclusions are not supported by the data presented.

The limitations are not clearly described

**Editorial and Data Presentation Modifications?**

Reviewer #1: Abstract

“Patients with uncomplicated malaria were recruited” should be Patients with uncomplicated vivax malaria …

“At the time of follow up” … please indicate the median follow-up time?

“… this may have implications for risk of drug induced haemolysis.” Many readers focus on the conclusions provided in the abstract. The conclusions as the are currently written are not helpful. Please indicate which implications do you have in mind? Are they good, bad, or neutral? You write in the discussion “in patients presenting with malaria the prevalence of severe and intermediate G6PD deficiency is considerably lower than that predicted from cross sectional surveys” which frequently guide prescription practices. Your study suggests that these prescription practices may be overly cautious. Please dare to be controversial.

Reviewer #2: (No Response)

Reviewer #3: (No Response)

**Summary and General Comments**

Reviewer #1: (No Response)

Reviewer #2: This is a very worthwhile study that carries important implications for assessing population-level P. vivax treatment risks. The manuscript is clearly presented, and the interpretation of results a good representation of the presented data.

Patients’ genetic variants were not characterised. If available, this would have been valuable additional data to help interpret the observations: authors could perhaps mention this in their discussion. It matters because the global diversity of G6PD mutations means that the trend observed in this study from these 3 sites may not be replicated everywhere. E.g. would this same difference be visible in patients with the Mediterranean mutation? Or is the effect less pronounced in patients where even young RBCs have low G6PD activity?

Minor comments:

Line 50: “replacing”: implies direct causation, suggest “leading to replacement of…”

Lines 87-90: for clarity, suggest listing references mid-sentence 

Line 112: not sure S2 table is right one to link to here?

Line 164: day 7 testing not previously mentioned; Methods implies all testing was on day 0.

Line 170: typo: “Norther Territory”

Line 269: “support previous model ?recommendations”?

Line 286 and 301: check wording

Reviewer #3: In this article the authors present very interesting data on the difference in G6PD activity in individuals when malaria positive versus when malaria negative. The data is interesting and the sheer number of cases for which they have this matched data could be useful. There are however some deficiencies in the data that need to be clarified and perhaps some issues in the manner in which the data is presented that should be resolved. The presentation of the statistics seems to be somewhat leading and possibly not necessarily supported by the data presented.

Major limitation of the study design: 

The study recruited the patients as malaria positive and then conducted a follow-up visit with a wide ranging period of time between the two visits. The study does not have a control arm of healthy individuals at the time of recruitment and time of follow-up. This is significant because all of the significant changes in G6PD status arise from the one site, Bangladesh, which noticeably (i) had the widest time gap between malaria and follow-up visits and (ii) was the only site to use the Randox assay and (iii) it is the one site where “ In Bangladesh participants were chosen to include the fraction with lowest G6PD activities during enrolment” so these will have had most samples that were on or close to the G6PD deficient/intermediate/normal thresholds. A lot of the differences may arise from slight differences in the Randox assay lots, the individuals/details of how the tests were run or differences in how the specimens were handled which would not be identified by the test controls. The AMM from the follow-up visits were used to determine the percent activity across both malaria and follow-up visits, so any bias resulting for the time lag would accentuate those observed. Without a control arm this cannot be resolved. These limitations should be specifically called out as they represent a significant limitation to the analysis. 

Noticeably the one site, Ethiopia where only P.vivax cases were collected showed the median G6PD activity increases in the follow-up. Which leads to another major limitation with regards to how the data is presented. The authors make their claims based on combined P.vivax and P.falciparum infections while their discussion is only relevant to P.vivax infections. It is surprising that the authors do not present the data disaggregated by species. Specially as the inherent (and not mentioned) assumption that both species have the same hematological effects is not addressed. The 5 samples that are G6PD deficient in the follow-up and not during the malaria infection, were all from Bangladesh and presumably (based on the discussion) in P.falciparum infections. 

Specifically:

Major limitations: 

1. The issue that there is no control arm for in this study should be clearly called out as it is a major limitation given the reasons mentioned above

2. The authors should present the change in G6PD activity disaggregated by species, P.falciparum and P.vivax and discuss the hemolytic risks according to the P.vivax results.

a. If there are differences observed here they should be discussed and the relevance to hemolytic risk also updated accordingly

3. In the abstract and discussion the authors should clarify that the significant drop in G6PD activity leading to different classifications almost entirely arise from the Bangladesh data (which has the above mentioned biases and limitations) and not from all three sites. The discussion should explain why the trend is only observed at one of the three sites, and specifically not observed in the site with only P.vivax infections (Ethiopia). The way the data is presented is misleading in that it suggests that this an observation across all sites.

4. Abstract lines 61 to 63: “67% (4/6) of deficient and 89% (36/41) of…” is an example of misleading statistics. Most of these cases (the 4 and 36) arise from the Bangladesh study in which all except one deficient and 4 intermediates were normal and so they could only go one way in the follow up visit. The Ethiopia and Indonesia study included all normal by FST so again this sample set was bias to only have switches towards intermediate or deficient cases. 

5. Author summary lines 40-42: “ …85% (40 / 47) of individuals ineligible for standard treatment during follow up, were eligible for treatment during malaria.” This is again misleading (assuming all the other issues are OK) since actually the 36 with 30-70% G6PD activity are eligible in most countries to 14 day primaquine. 

6. As per the above statement in the discussion lines 279-280, 70% is not the threshold for withholding primaquine, only those with G6PD activity below 30% would have primaquine withheld according to WHO guidelines. 

7. Line 282: “All three participants with lower G6PD activities during malaria compared to follow up,…” this seems to be misleading since their supplemental data and their Ethiopia data (Figure 1) which is relevant since these are all P.vivax cases there are than 3 cases lower G6PD activities compared to follow up.

PLOS authors have the option to publish the peer review history of their article (what does this mean?). If published, this will include your full peer review and any attached files.

Reviewer #1: No

Reviewer #2: No

Reviewer #3: No
---

## [Decision Letter · Decision Letter 1]

18 Mar 2022

Dear Dr Ley,

Thank you very much for submitting your manuscript "Variation in Glucose-6-Phosphate Dehydrogenase activity following acute malaria" for consideration at PLOS Neglected Tropical Diseases. As with all papers reviewed by the journal, your manuscript was reviewed by members of the editorial board and by several independent reviewers. The reviewers appreciated the attention to an important topic. Based on the reviews, we are likely to accept this manuscript for publication, providing that you modify the manuscript according to the review recommendations. 

Dear Authors,

Reviewer 3 lists several limitations of this study Please critically discuss.

BW

Hans-Peter Fuehrer

Sincerely,

Hans-Peter Fuehrer

Deputy Editor

Hans-Peter Fuehrer

Deputy Editor

Dear Authors,

Reviewer 3 lists several limitations of this study Please critically discuss.

BW

Hans-Peter Fuehrer

Reviewer's Responses to Questions

**Key Review Criteria Required for Acceptance?**

**Methods**

-Are the objectives of the study clearly articulated with a clear testable hypothesis stated?

-Is the study design appropriate to address the stated objectives?

-Is the population clearly described and appropriate for the hypothesis being tested?

-Is the sample size sufficient to ensure adequate power to address the hypothesis being tested?

-Were correct statistical analysis used to support conclusions?

-Are there concerns about ethical or regulatory requirements being met?

Reviewer #1: (No Response)

Reviewer #2: The methods are clearly presented.

Reviewer #3: (No Response)

**Results**

-Does the analysis presented match the analysis plan?

-Are the results clearly and completely presented?

-Are the figures (Tables, Images) of sufficient quality for clarity?

Reviewer #1: (No Response)

Reviewer #2: Results and associated figures are clear

Reviewer #3: (No Response)

**Conclusions**

-Are the conclusions supported by the data presented?

-Are the limitations of analysis clearly described?

-Do the authors discuss how these data can be helpful to advance our understanding of the topic under study?

-Is public health relevance addressed?

Reviewer #1: (No Response)

Reviewer #2: The conclusions correspond well to the results described and reflect their public health significance. The authors have added substantially to their discussion of the study limitations which was needed. These detailed limitations also now lay out for readers the challenge to carry out robust follow-on studies using prospective study designs in diverse epidemiological settings.

Reviewer #3: (No Response)

**Editorial and Data Presentation Modifications?**

Reviewer #1: (No Response)

Reviewer #2: Modification made to Line 203-204: this is now less clear, suggest simplifying to “10.4% higher during follow-up compared to at the time of acute malaria infection”

Reviewer #3: (No Response)

**Summary and General Comments**

Reviewer #1: The authors have addressed all of my suggestions adequately.

Reviewer #2: The authors have significantly revised the manuscript to address the previous reviews, which has improved it; the paper will make a valuable contribution to the literature with potentially important public health implications.

Reviewer #3: The authors have tried to address many of the issues raised by the reviewers. Several concerns still remain.

1. Bangladesh data set: There are several concerns with this data set and how it is presented. The follow-up data set has a median (and mean) G6PD value approximately 2.5 U/gHb lower than the acute malaria set and almost 2.0 U/g Hb lower than that of 100% normal used to determine the percent change in activity. In contrast the malaria and healthy G6PD medians are within 0.5 U/g Hb of each other and equally close to that of the authors 100% normal. This has several implications: (i) the Bangladesh data set is an outlier compared to the other two, and extremely unusual, (ii) using a 100% normal G6PD activity level that is 2 U/gHb or 40% greater than the population median does not make, by definition, any sense and will only exacerbate any difference with the acute malaria cases where the median was above the 100% normal, (iii) the bias generated by this will significantly and disproportionately affect any pooled analysis by species as presented by the authors in Table 2. 

2. Bangladesh data set: The authors only included G6PD deficient cases identified during the follow-up visit, and no deficient cases identified during the acute malaria phase, so inherently any analysis of the deficient cases will be biased and misleading. As the authors also point out, none of these were confirmed by sequencing, so it is just as likely that these are an artefact of the low G6PD values generated by the reference assay during this period. The controls seem to work well so possibly there was a specimen integrity issue. An indication that this sample set is strange is the data in supplementary table 5 with more “intermediate” males than females. 

3. Indonesia and Ethiopia: Given the outlier characteristics of the Bangladesh data set it is worth (and even in absence of that) looking at the G6PD levels per species per site and Ethiopia and Indonesia pooled. When this is done, at least by median G6PD values there is no significant difference in G6PD values between acute and follow-up visits, maybe a slight decrease for Pf (in Indonesia) and slight increase for Pv (in BOTH Indonesia and Ethiopia). By presenting the data the way it is presented in Table 2, rather than by species by site, as it would typically be done in this kind of analysis, this is not shown. 

4. Indonesia and Ethiopia: When only looking at these two data sets, the only changes in status is for 6 individuals, 3 of which are males so actually no change in status, and all males and females would have received primaquine. These changes in status maybe attributed to variation in reference testing

5. There is no biological or pharmacological sense in pooling the analysis across both species. Pv infects almost primarily reticulocytes and at lower parasite densities compared to Pf which infects a broader range of red blood cells and at higher parasite density. These differences are likely to have an impact on the relative effect of infection on G6PD activity levels, which their data (Indonesia and Ethiopia) actually seems to show. Notably (even in Bangladesh) all clinically relevant changes in G6PD activity are in Pf. 

6. None of the studies included deficient cases in the acute phase of malaria which was when the study participants were recruited. So any analysis based on these classifications is inherently biased. Changes in G6PD activity per site per species across all study participants and the statistical significance should be shown. And possibly pooled for Ethiopia and Indonesia.

PLOS authors have the option to publish the peer review history of their article (what does this mean?). If published, this will include your full peer review and any attached files.

Reviewer #1: No

Reviewer #2: No

Reviewer #3: No

Figure Files:

Data Requirements:

Reproducibility:

References

---

## [Editor Report · Decision Letter 2]

8 Apr 2022

Dear Dr Ley,

We are pleased to inform you that your manuscript 'Variation in Glucose-6-Phosphate Dehydrogenase activity following acute malaria' has been provisionally accepted for publication in PLOS Neglected Tropical Diseases.

Best regards,

Hans-Peter Fuehrer

Deputy Editor

Hans-Peter Fuehrer

Deputy Editor

---

## [Editor Report · Acceptance letter]

26 Apr 2022

Dear Dr Ley,

We are delighted to inform you that your manuscript, "Variation in Glucose-6-Phosphate Dehydrogenase activity following acute malaria," has been formally accepted for publication in PLOS Neglected Tropical Diseases.

Best regards,

Shaden Kamhawi

co-Editor-in-Chief

Paul Brindley

co-Editor-in-Chief
